# Patterns and Relevance of Langerhans Islet Invasion in Pancreatic Cancer

**DOI:** 10.3390/cancers13020249

**Published:** 2021-01-11

**Authors:** Ruediger Goess, Ayse Ceren Mutgan, Umut Çalışan, Yusuf Ceyhun Erdoğan, Lei Ren, Carsten Jäger, Okan Safak, Pavel Stupakov, Rouzanna Istvanffy, Helmut Friess, Güralp O. Ceyhan, Ihsan Ekin Demir

**Affiliations:** 1Department of Surgery, Klinikum Rechts der Isar, Technical University of Munich, Ismaninger Str. 22, D-81675 Munich, Germany; ruediger.goess@tum.de (R.G.); ayse.mutgan@medunigraz.at (A.C.M.); calisanumut@gmail.com (U.Ç.); yceyhun@icloud.com (Y.C.E.); renlei0524@gmail.com (L.R.); Carsten.Jaeger@mri.tum.de (C.J.); okan.safak@tum.de (O.S.); pavel.stupakov@tum.de (P.S.); rouzanna.istvanffy@tum.de (R.I.); helmut.friess@tum.de (H.F.); guralp.ceyhan@acibadem.com (G.O.C.); 2Department of General Surgery (Gastrointestinal Surgery), The Affiliated Hospital of Southwest Medical University, Luzhou 646000, China; 3Department of General Surgery, HPB-Unit, School of Medicine, Acibadem Mehmet Ali Aydinlar University, Istanbul 34684, Turkey; 4German Cancer Consortium (DKTK), Partner Site Munich, D-81675 Munich, Germany; 5CRC 1321 Modelling and Targeting Pancreatic Cancer, D-81675 Munich, Germany

**Keywords:** Langerhans islet, islet invasion, pancreatic tissue destruction, diabetes mellitus, survival, pancreatic cancer

## Abstract

**Simple Summary:**

The pathogenesis of pancreatic cancer-associated diabetes mellitus is poorly understood. We analyzed tumor infiltration into Langerhans islets and characterized it systematically for the first time, identifying four different main patterns of islet invasion. In a cohort of 68 pancreatic ductal adenocarcinoma (PDAC) patients, these islet invasion patterns were not related to occurrence of diabetes mellitus. However, severe islet invasion was associated with worsened overall survival.

**Abstract:**

**Background:** Pancreatic cancer-associated diabetes mellitus (PC-DM) is present in most patients with pancreatic cancer, but its pathogenesis remains poorly understood. Therefore, we aimed to characterize tumor infiltration in Langerhans islets in pancreatic cancer and determine its clinical relevance. Methods: Langerhans islet invasion was systematically analyzed in 68 patients with pancreatic ductal adenocarcinoma (PDAC) using histopathological examination and 3D in vitro migration assays were performed to assess chemoattraction of pancreatic cancer cells to islet cells. Results: Langerhans islet invasion was present in all patients. We found four different patterns of islet invasion: (Type I) peri-insular invasion with tumor cells directly touching the boundary, but not penetrating the islet; (Type II) endo-insular invasion with tumor cells inside the round islet; (Type III) distorted islet structure with complete loss of the round islet morphology; and (Type IV) adjacent cancer and islet cells with solitary islet cells encountered adjacent to cancer cells. Pancreatic cancer cells did not exhibit any chemoattraction to islet cells in 3D assays in vitro. Further, there was no clinical correlation of islet invasion using the novel Islet Invasion Severity Score (IISS), which includes all invasion patterns with the occurrence of diabetes mellitus. However, Type IV islet invasion was related to worsened overall survival in our cohort. **Conclusions:** We systematically analyzed, for the first time, islet invasion in human pancreatic cancer. Four different main patterns of islet invasion were identified. Diabetes mellitus was not related to islet invasion. However, more research on this prevailing feature of pancreatic cancer is needed to better understand underlying principles.

## 1. Introduction

To improve the prognosis of pancreatic cancer, early detection is urgently needed. Diabetes mellitus or hyperglycemia occurs in up to 80% of all pancreatic ductal adenocarcinoma (PDAC) patients, which usually emerges 2–3 years before the diagnosis of cancer [1,2,3]. This time period offers a perspective for using new-onset diabetes as a strategy for early detection [4,5]. Yet despite extensive previous research, it is still controversial whether diabetes mellitus should be seen as a risk for or a consequence of pancreatic cancer [6,7,8]. There are different hypotheses for the emergence of diabetes mellitus in pancreatic cancer: any form of stress including cancer development can unmask pre-existing type 2 diabetes. Further, as a consequence of tumor cachexia caused by a dysmetabolic state, diabetes can occur, which is seen similarly in several types of cancer [9]. Growing evidence in clinical and laboratory research supports the hypothesis that diabetes appears as a paraneoplastic phenomenon [10,11,12,13,14]. Furthermore, destruction of the pancreatic tissue containing the islets of Langerhans due to infiltrating tumor cells is a further possible explanation for pancreatic cancer-induced diabetes.

The islets of Langerhans embedded within acinar tissue are composed of four types of endocrine cells: the α- and β-cells, which regulate the glucose metabolism through secretion of glucagon and insulin, and the PP- and δ-cells that stimulate the secretory properties of other pancreatic cell types by producing pancreatic polypeptide and somatostatin [15]. However, infiltration of the endocrine pancreas tissue by tumor cells has not yet been studied in detail. There is no clear evidence that cancer-induced destruction of the Langerhans islets is causing diabetes mellitus or has any impact on the survival of PDAC patients.

Accordingly, we conducted this study to systematically analyze invasion patterns of the Langerhans islets in PDAC. First, we identified different patterns of islet invasion using double-immunolabeled resected human pancreatic cancer tissue from patients who underwent resection for PDAC. To investigate the role of endocrine/exocrine-related crosstalk for islet invasion, we performed 3D in vitro migration assays with pancreatic cancer cells exposed to Langerhans islets. Furthermore, we analyzed the impact of islet invasion patterns on diabetes mellitus and survival in 68 patients from our cohort.

## 2. Materials and Methods 

### 2.1. Patients and Histological Specimens

Human pancreatic cancer tissue specimens were collected from 68 patients diagnosed with PDAC during tumor resection in our institution. Informed written consent for tissue collection was provided by all patients. Follow-up data were obtained during regular hospital or outpatient visit or with follow-up calls. For the survival analysis, we had to exclude 8 patients due to metastatic lesions found during resection (M1 in liver or peritoneum), 2 patients receiving neoadjuvant therapy and 1 patient dying due to surgical complications. The demographic data and patient characteristics are summarized in Table 1. The collection of human specimens was approved by the ethics committee of the Technical University Munich on 12 November 2007 (approval no. 1926/07 with amendment 118/19s 11/20).

### 2.2. Cells Lines

The human PDAC cell line SU86.86 was purchased from ATCC. The mouse pancreatic cancer cells were derived from the KPC (LSL-Kras^G12D/+^; LSL-Trp53^R172H/+^; Pdx-1-Cre) mouse model. Pancreatic islet cells were freshly isolated from mice (C57BL/6 strain from Charles River Laboratories International Inc., Wilmington, MA, USA). After digestion of the pancreas with Collagenase P (Roche Diagnostics, Rotkreuz, Switzerland), the identifiable islets were collected under an SMZ 1500 stereomicroscope (Nikon, Düsseldorf, Germany) and used for the subsequent 3D migration assay. All cells were cultured with RPMI-1640 medium (Sigma-Aldrich, St.Louis, MO, USA) with supplementation of 10% fetal calf serum (FCS) and 1% Penicillin-Streptomycin.

### 2.3. Immunofluorescence

We used immunofluorescence double-staining to systematically analyze the collected PDAC specimens. Consecutive 3 μm sections from formalin-fixed and paraffin-embedded pancreatic tissue were incubated overnight with the corresponding antibodies (CK19-Anti-KRT19, rabbit, Sigma-Aldrich, Munich, Germany, 1:400; Mucin1-Anti-MUC1, mouse, Santa Cruz Biontechnology, Dallas, TX, USA, 1:400; Insulin-Anti-Insulin, mouse, HyTest Ltd., Turku, Finland, 1:500; and Glucagon-Anti-glucagon, rabbit, Cell Signaling Technology, Danvers, MA, USA, 1:400) after antigen retrieval via incubation in citrate buffer in a humid chamber at 4 °C. Finally, staining was detected with fluorescein-conjugated secondary antibodies (Alexa Fluor 488 and 594 antibodies, Life Technologies, Carlsbad, CA, USA, 1:300). Nuclei were stained using DAPI. Islet invasion patterns were analyzed by two independent observers (U.C. and Y.C.E) after imaging using the BioRevo BZ-9000 platform (Keyence, Osaka, Japan).

### 2.4. Islet Invasion Severity Score

For the analysis of islet invasion, we established the *Islet Invasion Severity Score (IISS)* to determine the severity of islet invasion. For this purpose, we categorized all islets of each patient into the different invasion patterns and calculated a score in the following standardized formula: individual IISS = 1 × (*healthy islets %*) + 2 × (*peri-insular invasion* %) + 3 × (*endo-insular invasion* %) + 4 × (*distorted islet structure* %) + 5 × (*adjacent cancer and islet cells* %). The score ranged from 100 to 500. We suppose that a higher IISS reveals a greater severity of islet invasion in each patient. The severity of islet infiltration increases from *peri-insular invasion*-type to *endo-insular invasion*-type to *distorted islet*-type. The most severe invasion of islets was observed in the *adjacent cancer and islet cells* type.

### 2.5. 3D Migration Assay 

We performed 3D migration assays to investigate the chemoattractive potential of pancreatic islet cells on pancreatic cancer cells. Here, we used the SU86.86 cell line and KPC cells and freshly isolated murine pancreatic islets. A total of 5 × 10^4^ cancer cells were resuspended in an ECM gel drop (Sigma-Aldrich, Munich, Germany) and placed on a 3.5 cm culture dish at the distance of 1 mm from an ECM gel drop with 5 pancreatic islets. As negative control, an empty drop was pipetted on the opposite side and the drops were connected via ECM gel “bridges” to generate the chemoattractive gradient. Migration was analyzed via digital time-lapse microscopy over a period of 24 h after 12 h of incubation, as described previously [16,17].

### 2.6. Islet Cell Mass Quantification 

To analyze the influence of the Langerhans islet mass on diabetes mellitus, we measured the area of β-cells for each patient. Therefore, we used the insulin immunofluorescence staining of 52 patients of our cohort as described above. We measured the area occupied by the immunostained pixels with insulin as the β-cell area on each slide using the threshold function of Image J (v1.53e, National Institute of Health, Bethesda, MA, USA) and compared it to the full area of each slide. The portion of the β-cell area to the full slide was classified as the mass of Langerhans islets.

### 2.7. Statistical Analysis

Graph Pad Prism 8 (GraphPad Software, California, CA, USA) and IBM SPSS Statistics v26 (New York, NY, USA) were used for statistical analyses. All data are expressed as mean ± SEM. Survival analysis was performed with the log-rank test and depicted as Kaplan–Meier curves. Comparing two groups in the 3D migration assay, we used the Mann–Whitney U test. All tests were two-sided, and a *p* value < 0.05 was considered to indicate statistical significance. All experiments were repeated at least three times independently.

## 3. Results

### 3.1. Langerhans Islet Invasion in Human PDAC Exhibits Four Different Morphological Patterns of Severity

We analyzed the infiltration of pancreatic cancer cells into pancreatic islets via double-immunolabeling of human pancreatic cancer tissues with insulin or glucagon and the pancreatic cancer cell marker cytokeratin 19 (CK-19) or Mucin-1 (MUC1). According to the infiltrated structures and damage on the islet architecture, we stratified “islet invasion” in the four following categories: (I) *peri-insular invasion* defined as tumor cells directly touching the round islet boundary without penetrating the islet (Figure 1A), (II) *endo-insular invasion* with tumor cells found inside the round islet, but still respecting the shape of the islet (Figure 1B), (III) *distorted islet structure* as cancer infiltration into the islet with distortion of the round islet morphology, yet still reminiscent of an islet (Figure 1C) and, finally, (IV) *adjacent cancer and islet cells*, defined as totally destroyed islets with consequently solitary islet cells adjacent to cancer cells (Figure 1D).

We found “islet invasion” in all sixty-eight analyzed PDAC patients. Eleven patients had only one specific pattern of islet invasion (8 patients showed only *adjacent cancer and islet cell* invasion, and 3 patients only *endo-insular invasion*). The remaining fifty-seven patients presented varying invasion patterns. The highest prevalence of islet invasion according to our classification was observed in the *adjacent cancer and islet cell* group with 96% (*n* = 65) of all patients, followed by the *endo-insular invasion* pattern in 81% (*n* = 55). In 68% (*n* = 46) of all patients, we observed *peri-insular invasion*, and in 68% (*n* = 46) the *distorted islet structure* pattern. To further study the frequency of islet invasion, we categorized every observable islet or islet-like structure in every patient into one of the groups or as healthy islets in the absence of invading tumor cells. On average, 50% of all islets were defined as *adjacent cancer and islet cells*, 22% as *endo-insular invasion*, 6% as *peri-insular invasion* and only 4% as *distorted islet structure*. We identified 18% of the islets without any sign of tumor infiltration and thus as healthy islets.

Interestingly, we also observed a subpopulation of cells that co-expressed insulin and CK-19 (Figure 1E). This co-expression of insulin and CK-19 was only present in 19 patients (28%) and was not related to the described islet invasion pattern.

### 3.2. Islet Invasion Is Not Associated with Diabetes Mellitus

Next, we investigated whether the occurrence of diabetes mellitus is related to the severity of islet invasion in each patient. We compared the 25 patients (37%) who presented with new-onset preoperative diabetes mellitus to 43 patients (63%) with no diabetes mellitus. In the analysis of the frequency of each invasion pattern between the diabetic (*n* = 25) and non-diabetic groups (*n* = 43), there was no significant difference in the distribution of these patterns (Figure 2A). The mean frequency of *peri-insular invasion* was 6.6 ± 1.3% (non-diabetic group) vs. 6.3 ± 1.9% (diabetic group), for *endo-insular invasion* 23.8 ± 3.8% (non-diabetic group) vs. 18.3 ± 5.3% (diabetic group), for *distorted islet structure* 3.0 ± 0.7% (non-diabetic group) vs. 4.8 ±1.3% (diabetic group) and for *adjacent cancer cell and islet cell* 46.7 ± 4.6% (non-diabetic group) vs. 54.3 ± 7.1% (diabetic group). Non cancer-infiltrated islets (“healthy islets”) were detected with a mean frequency of 19.4 ± 3.5% in the non-diabetic group vs. 16.3 ± 4.4% in the diabetic group.

In the comparison of the individual *Islet Invasion Severity Score* of patients with diabetes mellitus and without diabetes mellitus, we could not detect any significant difference (mean IISS non-diabetic group = 351 ± 14.5 vs. diabetic group = 375 ± 21.7). Therefore, we concluded that there was no relation between the severity of islet invasion and diabetes mellitus in our cohort.

To investigate the influence of the mass of Langerhans islets of each patients on the presence of diabetes mellitus, we performed islet cell mass quantification of 52 patients of our cohort (Figure 2B). The mean β-cell mass of patients with diabetes (*n* = 20) was 1.4 ± 0.4%, compared to 2.2 ± 0.5% in patients who did not present with diabetes mellitus (*n* = 32) (*p* = 0.55). Therefore, we assume that the mass of Langerhans islets of a patient does not impact on the occurrence of diabetes mellitus.

### 3.3. Impact of Islet Invasion Pattern on Patient Survival

Next, we correlated the survival data of our patients to the frequency and severity of the different islet invasion patterns. The mean survival of all 57 patients was 21 months. First, we analyzed survival in relation to each different invasion pattern separately. Therefore, each patient was categorized according to the mean frequency of each pattern into “*low invasion*” (from 0 to ≤ mean frequency) or “*high invasion*” (>mean frequency). There was no significant difference in patient survival depending on the severity of *peri-insular invasion* (median overall survival (mOS): low invasion 21.7 vs. high invasion 23.6 months, log rank 0.348), *distorted islet structure* (mOS: low invasion 22.8 vs. high invasion 25.6 months, log rank 0.776) and the *healthy islets* group (mOS: low healthy 25.3 vs. high healthy 21.7 months, log rank 0.440) (Figure 3A,B). We found significantly worse overall extended survival in patients with higher severity of *adjacent cancer and islet cell* (mOS: 27.9 vs. 15.9 months, log rank 0.025). Interestingly, patients with a lower severity of *endo-insular invasion* had a worse overall survival compared to patients with higher severity (mOS: low invasion 19.2 vs. high invasion 34.2 months, log rank 0.054).

To compare all different invasion patterns with regard to survival, we used our *Islet Invasion Severity Score* (*IISS*) with the same type of separation into the low-IISS and high-IISS groups. Here, we could not detect a significant difference in the comparison of the two groups, but it showed a trend towards better overall survival in the low-IISS group (Figure 3A). Median survival in the low-IISS group was 27.9 months compared with 18.3 months in the high-IISS group (log rank 0.127).

We also correlated the UICC cancer stages with the different islet invasion patterns. Therefore, we compared the mean IISS and low- and high-IISS groups of all patients with equal cancer stages (Table 2). There was no clinical significant correlation of the UICC cancer stages and the different islet invasion patterns.

The subgroup analysis of all patients with co-expression of insulin and CK-19 (*n* = 19) did not show a significant difference in median overall survival in comparison with patients who did not express both markers (*n* = 49). Patients with co-expression of insulin/CK-19 had a median overall survival of 19 months compared to 22 months without co-expression (log rank 0.537). There was also no correlation with different islet invasion patterns using the IISS and comparing both groups (*p* = 0.20) (Table 3).

We also correlated the type of resection performed with the IISS and survival in each patient (Table 4). Patients undergoing total pancreatectomy had significantly increased mean IISS (mIISS: 419) compared to patients receiving head resection (mIISS: 354) or distal pancreatectomy (mIISS: 327). Six of the 12 patients (50%) receiving total pancreatectomy had preoperative diabetes mellitus. This correlated with significantly shortened survival in patients after total pancreatectomy compared to patients after distal pancreatectomy or head resection, indicating that a high IISS is indicative of more aggressive and extended pancreatic cancer.

### 3.4. Islet Invasion Is Not Caused by the Endocrine/Exocrine Crosstalk

To further study whether pancreatic cancer cells are chemoattracted by endocrine cells in the initiation of islet invasion, we performed 3D migration assays. Cancer cells were exposed to freshly isolated murine islet cells on one side and an empty drop as negative control on the other side (Figure 4A). First, we used SU86.86 cells because of our previous data showing the invasive ability of these cells in a 3D environment [16]. After 24 h of migration, SU86.86 cells did not show any targeted movement toward the islet cells. The forward migration index (FMI) for the islet front was 0.83 compared to 0.79 in the back front (negative control) (Figure 4B). Then, we repeated the migration assay with mouse pancreatic cancer cells (KPC cells) to ensure confronting cells from the same species. The KPC mouse cancer cells also did not show any relevant chemoattraction towards the islet cells. The FMI for the islet front and the control back front was 0.11 (Figure 4C). Based on these observations, pancreatic cancer cells are obviously not chemoattracted by endocrine cells via paracrine secretion.

## 4. Discussion

In the present study, we performed the first systematic analysis of invasion of Langerhans islets in human pancreatic cancer. We identified four different main patterns of islet invasion with increasing severity of infiltration. In addition, we showed that islet invasion is not linked to the incidence of new-onset diabetes mellitus and has no impact on overall survival in resected PDAC patients.

Despite the fact that islet invasion has not yet been studied in pancreatic cancer, different observations in clinical practice negate a connection between islet destruction and PDAC-associated diabetes mellitus. After tumor resection for PDAC, diabetes mellitus often disappears or at least improves particularly in patients with new-onset diabetes [18,19]. These findings suggest, along with our results, that destruction of the endocrine pancreas cannot be one of the major causes of new-onset diabetes in PDAC. Here, limitations of our study have to be noticed, including lack of information on the exact starting point of diabetes and improvement after tumor resection. Furthermore, several studies propose that diabetes mellitus in pancreatic cancer more likely results from peripheral insulin resistance and/or impaired β-cell function rather than a decrease in or destruction of endocrine cells [20,21,22]. Genome-wide association studies have also shown a possible link between developing diabetes mellitus and pancreatic cancer through gene mutations in the hepatocyte nuclear factor 1 alpha (HNF1A) gene [23,24].

In the present study, we classified islet invasion based on the infiltration severity by cancer cells, which results in progressive destruction of the round islets. In our cohort, there was no correlation between these islet invasion patterns and diabetes mellitus. However, a possible explanation might be that even destructed or infiltrated islets are still functional and participate in the glucose metabolism. Pancreatic islets including β-cells seem to be less vulnerable, e.g., in chronic pancreatitis—another disease leading to secondary diabetes mellitus. Therefore, Schrader and colleagues investigated the tissue destruction of the endocrine pancreas in chronic pancreatitis and showed that despite reduced pancreatic volume and reduced β-cell area, there was a 10-fold increase in apoptotic acinar cells than endocrine cells [25]. As the glucose homeostasis regulated by islets cells reflects a critical issue to health (more than maintenance of exocrine pancreatic function), destruction by tumor infiltration seems to affect the exocrine cells earlier rather than the endocrine cells [26].

Next, we speculated whether infiltration and distortion of islets can be a sign of more aggressive disease, which would be in parallel with infiltration into lymph nodes, nerves, vessels or surrounding tissue according to the UICC tumor classifications [27]. Therefore, survival times of our cohort were studied in relation to different invasion patterns. We only identified significantly reduced survival in patients presenting with higher severity of the *adjacent cancer and islet cell* pattern. Here, the islets were completely destroyed by tumor cells. Noticeably, there was no correlation of overall survival and the severity of other islet invasion patterns.

The established *Islet Invasion Severity Score* was used for a more specific and weighted comparison in analogy with studies on neural invasion in cancer [28,29]. The barrier for tumor cells to infiltrate islet structures seems to be low, as it was present in every single patient of our cohort. This phenomenon could restrain a visible and remarkable impact of islet invasion on survival.

To understand cell–cell interactions with potential chemoattraction leading to invasion of islets, we performed 3D migration assays. Interestingly, tumor cells were not attracted to infiltrated islets by paracrine crosstalk. Pancreatic cancer cells may possibly infiltrate into (rather than be chemoattracted specifically by) islets, which supports the lacking correlation between survival and severity of invasion. Further studies have to include the complexity of the surrounding microenvironment, i.e., immune cells and stromal fibroblasts, which may also affect islet invasion.

## 5. Conclusions

In conclusion, cancer invasion of islets has been found in every patient of our cohort, highlighting a common feature in human pancreatic cancer. We defined, for the first time, a classification which can easily be used in ongoing research on this topic. The true impact of this type of cancer invasion on patients and the underlying principles for this type of expansion in human PDAC are still unknown. Therefore, further investigation on islet invasion is needed to understand and unravel this newly described mode of tumor infiltration.

## Figures and Tables

**Figure 1 cancers-13-00249-f001:**
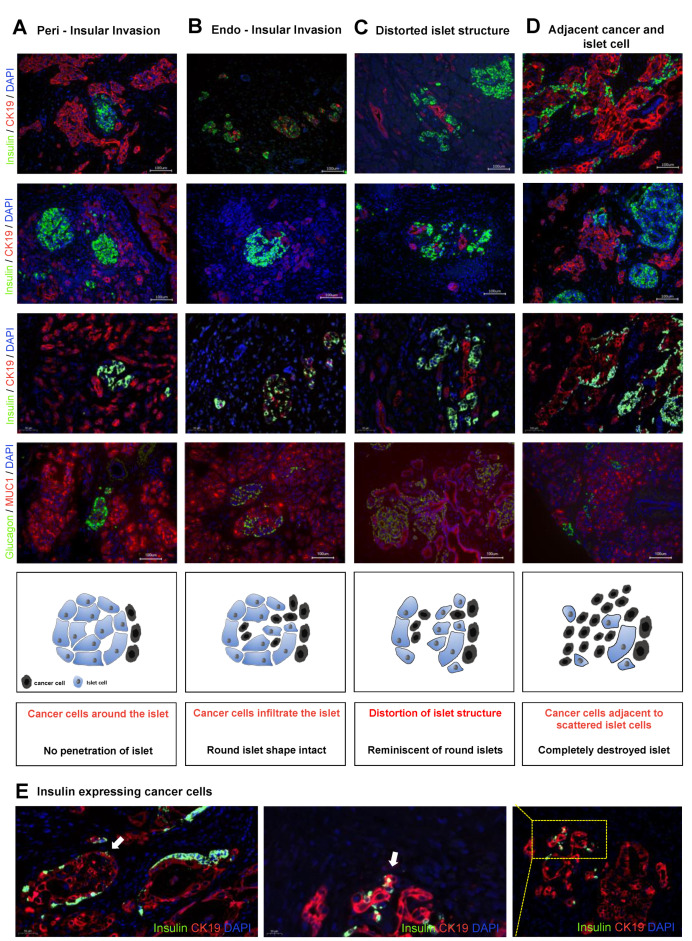
Classification of islet invasion pattern according to morphological appearance. Resected human pancreatic cancer tissues were double-immunolabeled with cytokeratin-19 (CK-19) or Mucin 1 (MUC1) for cancer cells and with insulin or glucagon for islet cells. We identified different islet invasion patterns shown each in three representative photomicrographs and an illustrated graphic: (**A**) *peri-insular invasion*: cancer cells around, but not penetrating, the islet. Islet has a preserved round structure; (**B**) *endo-insular invasion*: cancer cells infiltrate the islet structure without distorting the round shape of the islet; (**C**) *distorted islet structure*: complete loss of the round islet structure due to tumor infiltration, yet still reminiscent of an islet; and (**D**) *adjacent cancer and islet cell*: completely destroyed islets, and scattered endocrine cells are adjacent to cancer cells. (**E**) Insulin-expressing cancer cells: rare subtype, in which cancer cells exhibited insulin expression (white arrows).

**Figure 2 cancers-13-00249-f002:**
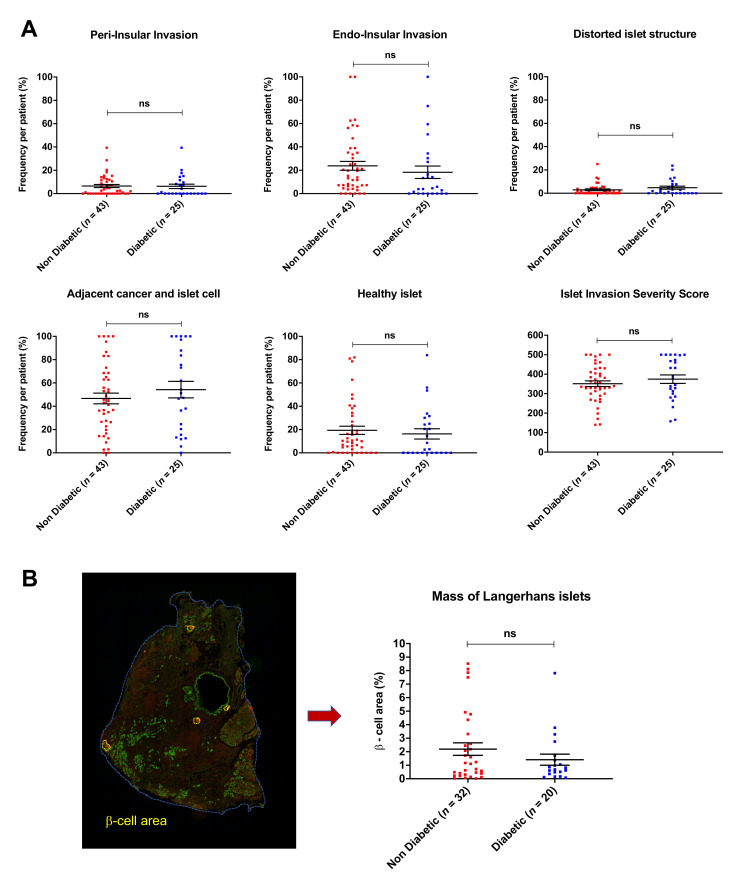
Impact of different islet invasion patterns and β-cell mass on diabetes mellitus. (**A**) We analyzed the correlation between diabetic (*n* = 25) and non-diabetic (*n* = 43) resected pancreatic ductal adenocarcinoma (PDAC) patients with regard to the different islet invasion patterns. All islets of three different specimens from each patient were categorized. There was no significant correlation between diabetes mellitus and the frequency of each pattern. Mann–Whitney U test was used to compare the results (ns = not significant). (**B**) Impact of mass of Langerhans islets on diabetes mellitus. Left side: representative photomicrograph of human pancreatic cancer tissues double-immunolabeled with cytokeratin-19 (green) and insulin (red). β-cell area surrounded in yellow.

**Figure 3 cancers-13-00249-f003:**
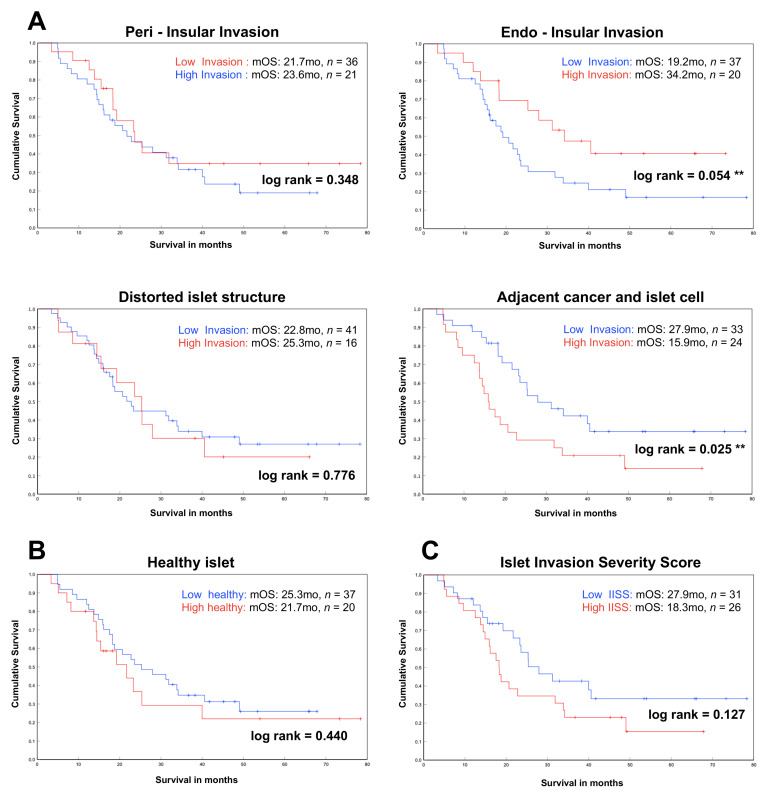
Impact of different islet invasion patterns on survival. (**A**) Impact of frequency of different invasion patterns on overall survival of patients (*n* = 57). Significantly prolonged survival was observed in patients with “high invasion” and the *endo-insular invasion* pattern. Patients with “low invasion” of the *adjacent cancer and islet cell* pattern showed significantly improved overall survival. (**B**) Impact of healthy islets on survival. (**C**) Impact on survival using the *Islet Invasion Severity Score* (IISS); analysis was performed using the Kaplan–Mayer method and the log rank test. Patients were categorized in each pattern as “high invasion”, as > mean, and “low invasion”, as ≤ mean, for analysis (mOS = median overall survival, mo = months, ** = significant result).

**Figure 4 cancers-13-00249-f004:**
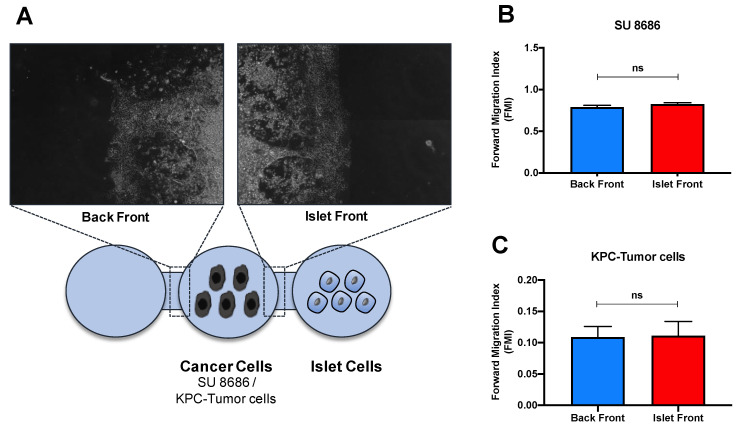
Impact of endocrine/exocrine cell crosstalk in cancer cell migration. (**A**) SU86.86 and KPC pancreatic cancer cells were exposed to freshly isolated pancreatic islet cells in a 3D migration assay and were analyzed for 24 h with time-lapse microscopy. (**B**,**C**) Forward migration index (FMI) was calculated by comparing the migration front of cancer cells toward islet cells with the migration of cancer cells toward the empty gel (back front). Mann–Whitney U test. ns = not significant.

**Table 1 cancers-13-00249-t001:** Demographic data and patient basic characteristics.

	All Patients	
	*n* = 68	
	*n*	%
Age (median, years)	66.0	
Sex		
Male	43	63.2
Female	25	36.8
Neoadjuvant Therapy	2	2.9
Type of surgery		
Head resection	42	61.8
Distal pancreatectomy	14	20.7
Total pancreatectomy	12	17.6
Tumor stage		
T1	0	0
T2	4	5.9
T3	54	79.4
T4	10	14.7
Lymph node		
N0	27	39.7
N1	20	29.4
N2	21	30.9
Grading		
1	4	5.9
2	36	52.9
3	28	41.2
Resection margin		
R0	40	58.8
R1	22	32.4
RX	6	8.8
UICC Classification (8th)		
I	2	2.9
IIA	24	35.3
IIB	25	36.8
III	9	13.2
IV **	8	11.8
Median survival (months)	20.7	(*n* = 57)
Diabetes mellitus		
Yes	25	36.8
No	43	63.2

** excluded from the survival analysis.

**Table 2 cancers-13-00249-t002:** Correlation of UICC cancer stage with islet invasion patterns and diabetes mellitus.

UICC	*n*	Diabetes (%)	No Diabetes (%)	IISS (Mean)	Low IISS	High IISS
IB	19	6 (32)	13 (68)	360	10	9
IIA	7	3 (43)	4 (57)	363	3	4
IIB	15	8 (53)	7 (47)	332	10	5
III	19	3 (16)	16 (84)	365	10	9
IV	8	5 (62)	3 (38)	397	3	5
total	68	25 (37)	43 (63)	360	36	32
		ns	ns	ns	ns	ns

IISS: Islet Invasion Severity Score; ns: not significant.

**Table 3 cancers-13-00249-t003:** Correlation of co-expression of insulin/CK-19 with survival and IISS.

Co-Expression Insulin/CK-19	*n* (%)	IISS (Mean)	mOS (Months)
Yes	19 (28)	385	19
No	49 (72)	350	22
		*p* = 0.20	log rank = 0.537

IISS: Islet Invasion Severity Score; mOS: median overall survival; Mann–Whitney U test.

**Table 4 cancers-13-00249-t004:** Correlation of resection type with IISS and survival.

Type of Resection	n (%)	IISS (Mean)	Low IISS	High IISS	mOS
Head resection	42 (61.8)	354	24	18	19
Total pancreatectomy	12 (17.6)	419 *	2	10	18 **
Distal pancreatectomy	14 (20.6)	327	10	4	34

* significant/Mann–Whitney U test; ** significant/log rank test. Islet Invasion Severity Score (IISS): head vs. total resection *p* = 0.043; total vs. distal pancreatectomy *p* = 0.011. Median overall survival in months (mOS): head vs. total resection log rank = 0.045; total vs. distal pancreatectomy log rank = 0.002; head vs. distal pancreatectomy log rank = 0.434.

## Data Availability

The data presented in this study are available on request from the corresponding author.

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
