# Peer review of "Patterns and Relevance of Langerhans Islet Invasion in Pancreatic Cancer"

_cancers, 2021, doi:10.3390/cancers13020249_

Round 1

Reviewer 1 Report

all concerns are addressed.

Reviewer 2 Report

The authors have nicely addressed all of my comments. No additional comments.

Reviewer 3 Report

The authors have submitted a much improved version of their manuscript, in which they are answering the reviewer's question right.

We have no additional comments and consider the latest version of the manuscript suitable for publication.

This manuscript is a resubmission of an earlier submission. The following is a list of the peer review reports and author responses from that submission.

Round 1

Reviewer 1 Report

In this manuscript the authors address the question whether invasive destruction of Islets in pancreatic cancer can underlie onset of diabetes seen in pancreatic cancer patients. The authors create a islet invasion score to assess this in human pancreatic cancer tissues and determine if a correlation between islet invasion and diabetes or islet invasion and overall survival exists. The article is clearly written and well-structured. Although this is theoretically negative data, it is answering an important question that will be of interest to others.

major comments:

  1. information is missing how patient tissues were fixed and prepared histologically
  2. The authors use SU86.86 cells to study if these cells are attracted to islet cells. This experiment is missing a control to show that SU86.86 are actually invasive cells (they are, but this is not referenced or shown) and are able in the experiment the authors are conducting to move (are the experimental conditions allowing for cell motility). It would be good if the authors can add a chemoattractant instead of islet cells and at least show that the system works.

minor comments:

        I have one minor point to ask if the authors can include a positive control  in figure 3.

Reviewer 2 Report

Ruediger Goess et al in this work described the relationship of Diabetes mellitus or hyperglycemia and pancreatic ductal adenocarcinoma (PDAC) and they identified four different patterns of islet invasion of human pancreatic cancer : (Type I-IV). They concluded that Diabetes mellitus was not related to islet invasion of PDAC cancer cells. However, the result from this study is not convincing and further evidence is needed. 

  1. Does the in vitro cell culture of human cancer cells SU8686 and primary mouse islet cells really mimic the in vivo? The experiment could not rule out the involvement of immune cells in the recruitment of cancer cells. In addition, the authors should provide evidence that the primary islet cells share the broad signature of in vivo islet cells. mouse cancer cell could be better than SU8686 when co-culturing with mouse islet cells.
  2. The authors need to provide the correlation of different islet invasion patters with the cancer stages, as well as the correlation of cancer stages with the diabetes mellitus.
  3. Different cancer cell markers and islet cells marker may be needed to confirm the result presented in Fig1.

Reviewer 3 Report

Review for cancers-930557-peer-review-v1

General Comments: Overall, very well written and presented manuscript. A few major comments:

1) Expand upon the likelihood of both the epithelia AND the islets having identical or similar driver mutations that therefore result in DM.

2) For the SU86.86 cells, specify the species of origin and if from mice, are they from C57BL6. This is of interest since using immunocompetent C57BL6 mice for islets.

3) In the statistical analysis, add tests used for 2 group comparisons.

4) Line 152-154 – for co-expression of CK-19 and insulin, see previous driver mutation comment.

5) How does the survival of those 14 patients co-expressing CK-19 and insulin relate to 1) the invasion groups (even though I know they come from them) and 2) those not co-expressing CK-19 and insulin?

More Specific Comments:

Abstract

1) Line 27-29 – Clarify here that the IISS is the same scoring system described just above this.

2) Line 30 – Move “systematically” to before “analyzed”

Introduction – None

Materials and Methods

1) Line 69 – Change “lesion” to “lesions”

2) Line 102 – Check spacing between 3. And D

Results

1) Line 203 – Change “confronted with” to “exposed to”

Discussion – None

Conclusions – None

Figures and Tables – None

Reviewer 4 Report

The authors performed a very interesting paper looking in the role of infiltrating pathology PDAC in front of the Langerhans's Islets invasion and their morphological changes. The authors proposed a very interesting score in order to validate their scopes. The concept work is well described and the results seem to support their conclusions. However, in our personal opinion some additional modification should be considered to increase power of manuscript. In particular, we would like to invite the authors to have a look in the following list of major and minor criticisms.

Major Comments:

  • Table 1 reports all the pathological findings of PDAC patients undergoing a surgical procedure, with curative intent. The authors used the older UICC classification for TNM and the UICC reclassification of PDAC patients (compliant with the eighth edition of the UICC manual) is mandatory. So far, the right indication of N status and tumor staging is too important.
  • Looking at the same table cited above, the authors did not report the type of resection performed in PDAC patients. This information could be useful in determining whether patients who underwent different surgical treatments showed different islet infiltrations depending on their presence in the surgical specimens (i.e. there are fewer islets in the pancreas head than in the tail).
  • Did the authors also perform total pancreatectomies in patients with DM-PDAC? If so, how many times?
  • The authors should report a table showing the glucose concentration in both groups of patients (with DM and without DM), before and after surgery.
  • The authors could extend the morphological analyzes of the islets to look for the mass of beta cells in the two different groups to verify whether PDAC infiltration affects DM status.
  • Looking at the in vitro experiment we would like to ask the authors if they have the option of using human-made islets of langerhans instead of those derived from mice.
  • Looking at the migration experiment, might the xenon cell model used not be appropriate? Furthermore, could the authors provide some data regarding the migration of some molecular factors (i.e. ELISA test)?

Minor comments:

  • Authors should better describe antigenic retrieval in order to perform dual immunofluorescence. No data on the chemicals used has been reported.
